



# Seasonal impact of submesoscale eddies on the ocean heat budget near the sea ice edge

Lily Greig[*1] and David Ferreira[2]

[1]Department of Mathematics, University of Reading
[2]Department of Meteorology, University of Reading

**Abstract.** Oceanic submesoscale mixed layer eddies (SMLEs), with horizontal scales of 0.1-10 km, are not captured in climate models. SMLEs energized in the marginal ice zone (MIZ) have been shown to be of importance to sea ice melt rates in summer and to sea ice transport through notably a dynamical coupling with sea ice. Here our focus is on the thermodynamical coupling, which has received comparatively little attention. We aim to quantify, for the first time, the impact of eddies on both sea ice
and the heat budget in the MIZ, contrasting different seasons and different background stratifications.

To this end, we set up SMLE-resolving simulations of the ocean mixed layer (ML) near the ice edge using the MITgcm, representing a lead or the MIZ. We isolate the effect of eddies by comparing 3D simulations with eddies to 2D (latitude-depth) simulations without eddies.

In summer (i.e., melting conditions) and regardless of the background stratification, SMLEs act as a heat pump from the
atmosphere over the open ocean to the sea ice. On average over a season, SMLEs triple the meridional heat transport to the ice covered region, increase melting over their meridional extent, and trigger a positive radiative feedback by increasing shortwave absorption over the thinner ice. These changes are in the range 20-60% for reasonable choices of shortwave forcing and initial ice thickness. In winter (i.e., freezing conditions), SMLEs have a relatively small impact on sea ice growth due to compensation between vertical and horizontal eddy heat transports. However, they reduce ML deepening by 80/50% in the open/ice-covered
ocean. Overall, our results reveal up to order one impacts of SMLEs on the heat and sea ice budgets in the MIZ, which will require the development of a SMLE parameterization tailored for polar regions.

## 1 Introduction

Sea ice is a critical component of the climate system. This is because of its high albedo and the way it mediates air-sea heat
and momentum exchanges in polar regions. Predicting the evolution of the sea ice cover in both the Arctic and Antarctic is extremely challenging due to a lack of understanding of the climate processes influencing it (IPCC, 2023; Stroeve et al., 2012). Ocean heat content and transport have been shown to be a major regulator of the position of the sea ice edge (Bitz et al., 2005;

---

[*]Now at: Centre for Environment, Fisheries & Aquaculture Science (CEFAS). Correspondence: lily.greig@cefas.gov.uk



Aylmer et al., 2024, and references therein). This oceanic influence can be direct from the ocean to the sea ice but can also be mediated by air-sea exchanges near the ice edge (Aylmer et al., 2022). Aylmer et al. (2022) show for instance that increased ocean heat transport toward the Arctic is associated with increased heat release equatorward of the ice edge and increased atmospheric energy transport toward the central ice pack. As the Arctic sea ice cover is becoming increasingly dominated by the Marginal Ice Zone (MIZ) – the region that lies between pack ice and open ocean (Vichi, 2022; Rolph et al., 2020; Frew et al., 2025), it is likely that such ocean-atmosphere-sea ice interactions will become increasingly important in regulating the sea ice extent.

Building knowledge of the oceanic mechanisms at play here is critical to making progress in our understanding (Docquier and Koenigk, 2021). Both the MIZ and leads (cracks in the sea ice cover) represent areas of strong lateral density gradients in the upper ocean due to spatial heterogeneity of buoyancy forcing to the ocean. These upper ocean fronts lead to the energisation of submesoscale processes (Swart et al., 2020; Horvat et al., 2016; Manucharyan and Thompson, 2017).

The oceanic submesoscale is typically defined by processes with length scales of the order of 100 metres to 10 kilometres, and time scales of days to weeks, i.e., Rossby number of order 1. These processes are associated with the weakly stratified ocean mixed layer (ML), and are also associated with Richardson numbers of order 1. These processes were first observed from space in the 1960s and, due to further observations since, are now regarded to be ubiquitous in the ocean (Munk et al., 2000). Amongst the submesoscale dynamics are submesoscale mixed layer eddies (SMLEs). These baroclinic SMLEs develop through the baroclinic instability in the ML (Haine and Marshall, 1998; Fox-Kemper et al., 2008). Due to their small scales, SMLEs are not resolved in global climate models, yet they are a potentially important process to take into account at global scales (Hewitt et al., 2022). SMLEs have the potential to set ML depth (MLD) and to influence the stratification of the surface ocean and air-sea interactions (Hosegood et al., 2006; Fox-Kemper et al., 2011; Thompson et al., 2016; McWilliams, 2016; Taylor and Thompson, 2023).

In the last decade or so, SMLEs in polar regions have received increasing attention. Observational campaigns initially sought to find evidence for the presence of submesoscale fronts and flows in polar regions, which were found in abundance (Timmermans et al., 2012; Timmermans and Winsor, 2013; Giddy et al., 2020; Swart et al., 2020). One focus of such campaigns has been to understand the interaction of SMLEs with their environment, which, critically, in polar regions includes interactions with sea ice. Evidence has been found that the seasonal cycle of submesoscale eddy heat fluxes and their interaction with winds is partly established by the seasonal cycle of the sea ice cover (Giddy et al., 2020; Swart et al., 2020).

Alongside these observational campaigns, numerical modelling studies have brought to light different mechanisms of SMLE-sea ice interaction. During the melt season, SMLEs have the potential to transport heat under ice and influence lateral melt rates. As a result of this interaction, the dependence of floe melt rates on floe size extends in the range 1-50 km (smaller floes, which have a larger perimeter for the same sea ice volume, melting faster), a much larger scale range than previously thought (Horvat et al., 2016; Horvat and Tziperman, 2018). Eddies, if energised in the MIZ, could trap sea ice floes and advect them into warmer waters, resulting in increased ice melt (Manucharyan and Thompson, 2017), although the details of the trapping mechanism remain debated (Gupta and Thompson, 2022). In leads, ice-ocean stresses were found to have a limited influence



on the eddy development, whilst another study found that in partially sea ice-covered regions ocean-ice friction could reduce eddy overturning by 40% (Cohanim et al., 2021; Shrestha and Manucharyan, 2022).

The impacts of SMLEs in polar regions are influenced by the background oceanic and atmospheric environmental conditions of MIZs (winds, ocean stratification, shortwave radiation reaching ocean surface) (Gupta and Thompson, 2022; Swart et al., 2020). These background conditions set the parameters that control SMLE fluxes (MLD, lateral density gradient) and therefore could control the eddy feedback onto these background properties too. For example, the eddy length scale is set by the ML Rossby radius of deformation, which scales with the ambient stratification (Thomas et al., 2008). The horizontal scale of SMLEs whose development is driven by brine release as sea ice forms within leads has been shown to scale directly with the salt input, also indicating the importance of surface forcing for the SMLE field (Matsumura and Hasumi, 2008).

       The prevalence and importance of SMLEs is expected to increase under global warming and with the increasing width of the Arctic MIZ (Manucharyan and Thompson, 2022). The impact of these eddies on air-sea heat fluxes remains largely unknown, as is the dependence of the magnitude of these impacts on sea ice and other climate variables that depend strongly on the season and background stratification.

In the present study, we isolate the thermodynamic impacts of eddies in polar regions, as it was less of a focus in most previous studies. Specifically, our aim is to understand the thermodynamic impact of SMLEs on sea ice, air-sea fluxes, and MLD in the vicinity of the sea ice edge (representing the MIZ or a lead). We also expand on previous work by contrasting how the impact of SMLEs is modulated by seasons (sea ice growth versus melting conditions), background stratification (Arctic versus Antarctic), and initial MLD. To this end, we compare 3D submesocale-resolving simulations with 2D (latitude-depth) simulations without eddies. This allows us to quantify eddy impacts through sea ice volume changes and heat budget terms, which we link to the eddy-induced overturning streamfunction (Fox-Kemper et al., 2008). We show that SMLEs have a leading order impact on the evolution of the ocean-ice state although the mechanisms are radically different between seasons. We also find that the background stratification is a key control in winter conditions but not in summer conditions.

       The structure of the paper is as follows. In section 2, the experimental set-up and the analysis methods used are described. Section 3 gives the results, of which there are four subsections, Arctic summer (3.1), Arctic winter (3.2), Antarctic (3.3), and sensitivity analysis (3.4). For Arctic summer, first, the development of SMLEs is described (section 3.1.1). Next in section 3.1.2 the influence of eddies on sea ice and air-sea fluxes is explored through heat budgets and through comparisons between 3D simulations with eddies and 2D simulations (no zonal variation) without eddies. In section 3.1.3 the eddy dynamics are described through the residual-mean theory framework. The same order is followed for Arctic winter in sections 3.2.1-3.2.3 and in 3.2.3 the eddy impact to MLD is investigated. The difference in eddy impact on sea ice, ML heat storage, air-sea flux and MLD depending on season and background stratification is discussed in section 4, where the key results are also summarised.

## 2   Materials and Methods

In this study, the Massachussetts Institute of Technology general circulation model (MITgcm) is employed (Marshall et al., 1997b, a). The model set-up is developed from and based on that used by Horvat et al. (2016). The domain is a hydrostatic





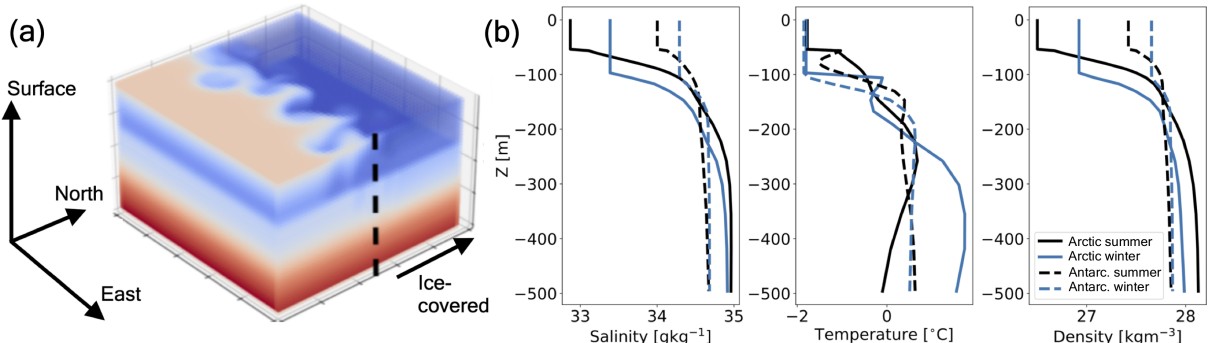

**Figure 1.** (a) Arctic summer set-up (temperature field), where the northern region is initially ice-covered and the southern region is ice-free.(b) initial salinity, temperature and density profiles for the four main experiments: Arctic/Antarctic (solid/dashed lines) and summer/winter (black/blue lines) conditions.

reentrant zonal channel on an f-plane and is 75 km in both horizontal directions with walls on northern and southern boundaries. The horizontal grid spacing is 250 m and the vertical grid spacing is 2.5 m over the top 75 m, increasing by 20% at each subsequent grid point to a total depth of 538 m. A thermodynamic sea ice package is used such that the sea ice advection by the ocean is switched off. In addition there is a free-slip boundary on the ice-ocean interface. The northern horizontal extent of the domain is initially covered with 1 m thick sea ice, and the southern half is open ocean. Further there is no wind forcing. Vertical mixing is represented by the K-profile parameterisation mixing scheme (Large et al., 1994). Under summer conditions, a Smagorinsky viscosity scheme is used, with the Smagorinsky scaling coefficient set to 2. In winter conditions, vertical mixing and vertical velocities are much more intense than in the summer due to the large surface buoyancy loss (e.g., sensible heat loss, brine rejection). This requires higher viscosity to maintain numerical stability. To avoid damping all motions, we employed a horizontal eddy viscosity of 50 $\mathrm{m^2\ s^{-1}}$ on the divergent component of the flow and of 1 $\mathrm{m^2\ s^{-1}}$ on the rotational component of the flow in winter simulation. In addition, a small horizontal diffusivity of 1 $\mathrm{m^2\ s^{-1}}$ is also used in winter simulations. An atmospheric boundary model evolves the atmospheric temperature in response to air-sea/air-ice fluxes (Deremble et al., 2013). There is no evaporation or precipitation between ocean and atmosphere.

A 2-dimensional version ($y - z$ only but otherwise identical to the 3D simulation) of this configuration is also set up. In this configuration named "no eddies" hereafter, SMLE are unable to develop. Comparison between the 2d and 3d simulations provides measures of the impact of eddies (see section 2.1).

Four main experiments are carried out representing combinations of Arctic/Antarctic conditions and summer/winter conditions. The Antarctic and Arcic conditions are distinguished through the choices of the initial temperature and salinity profiles. These are taken from Good et al. (2013), the EN4.2.1 ocean data near the ice edge. The single 2016 profiles at (84°N, 0°E) were taken for Arctic summer (July) and (83°N, 0°E) for winter (January). Antarctic initial temperature and salinity profiles were taken near the ice edge at (63 °S, 0°E) for the winter ((July) and (69 °S, 0°E) for the summer (January). The ML temperature





in all initial profiles is set to freezing point and for the winter/summer the initial MLD is 100/55 m. Initial temperature and salinity profiles are shown in Fig. 1 and are horizontally uniform over the whole domain.

The atmospheric forcing and initial MLD values are (loosely) based on the European Centre for Medium-Range Weather Forecasts' atmospheric reanalysis version 5 (ERA5) data from near the ice edge at 0°E in 2016. The atmospheric temperature $T_a$ is prescribed at -10 °C. A uniform shortwave forcing ($F_{SW,d}$) of 210 W m$^{-2}$ and longwave forcing ($F_{LW,d}$) of 220 W m$^{-2}$ are applied at the ocean surface. In winter, we use $T_a$=-5 °C, $F_{SW,d}$=50 W m$^{-2}$, and $F_{LW,d}$=175 W m$^{-2}$, which can be considered gentle winter conditions.

$Q_{net}$ (for the open ocean) is thus given by:

$$Q_{net} = F_{LW,d} - \epsilon \sigma T^4 + F_{SH} + F_{SW,d}, \tag{1}$$

where $F_{LW,d}$ is the downwelling longwave radiation, $\epsilon \sigma T^4$ is the outgoing longwave radiation, $\epsilon$ is the ocean emissivity, $\sigma$ is the Stefan-Boltzmann constant, and the net longwave radiation $F_{LWnet} = F_{LW,d} - \epsilon \sigma T^4$. $F_{SH}$ is the sensible turbulent heat flux and $F_{SW,d}$ is the downwelling shortwave radiation.

All simulations are run for 110 days to roughly match the length of the season. To allow instabilities to develop in the 3D model, the initial temperature was seeded with small-amplitude white noise between 0 and 0.05 °C, so that the mean initial temperature in the mixed layer is slightly (+0.025 °C) above freezing. Results were found to not be sensitive to the magnitude of the noise in the initial temperature field (tests were carried out for noise ± 0.0125-0.8 °C).

## 2.1 Residual-mean framework

The time-mean Eulerian Overturning Streamfunction (EOS) is calculated via zonal integration at a fixed height of the meridional velocity field:

$$\overline{\psi}(y,z) = -L_x \int_{H_d}^{z} \overline{v}(y,z')\mathrm{d}z', \tag{2}$$

where $H_d$ indicates the ocean bottom, $L_x$ is the zonal extent of the domain, and $\overline{v}$ is the time and zonally-averaged meridional velocity field, which removes eddy effects.

Following Abernathey et al. (2011), the isopycnal streamfunction is defined as:

$$\psi_{iso}(y,\rho) = \int_X \int_{\rho_d}^{\rho} (vh)\mathrm{d}\rho'\mathrm{d}x, \tag{3}$$

where $h = \frac{-\partial z}{\partial \rho}$ is a density layer thickness, $\rho_d$ is the density of the deepest isopycnal, and $\rho'$ is a dummy variable of integration. To calculate $\psi_{iso}$, the *layers* package in MITgcm is used. It calculates the layer transport $vH_\rho(x,y,\rho,t)$ online (at every time step), where $H_\rho(x,y,\rho,t)$ is the thickness of a density layer bin in metres (time averaging can be done later on). These density layer bins are chosen by the user at the start of the simulation. The full range of density bins chosen must cover the full range



of ocean densities included in the simulation as well as their variation in time. $\psi_{iso}(y, \rho)$ in Eqn. (3) is calculated by taking the cumulative sum of $\overline{vH_\rho}(y, \rho)$ (the sum was taken from the deepest isopycnal upwards). In this way, the meridional velocity $v$ is integrated with respect to isopycnal layers rather than fixed vertical levels as in $\overline{\psi}(y, z)$. Ideally, $\psi_{iso}(y, \rho)$ should be robust to the choice of density bins used in its calculation. Here, $\psi_{iso}(y, \rho)$ was found to be robust to the number of density layers when it is 92 or higher (tests were performed with 184 layers, and a 7% or less L2 error was found for all four of the summer and winter experiments).

$\psi_{iso}(y, \rho)$ in Eqn. (3) is a function of density, not depth. Therefore it is convenient to remap it to height coordinates in order to compare it to $\overline{\psi}$. Following the method of (Wolfe and Cessi, 2015), the remapping $\psi_{iso}(y, \rho) = \psi_{iso}[y, \overline{\rho}(y, z)]$ is used. In practice, first, the cumulative sum of the time and zonally averaged layer thicknesses $\overline{H_\rho}(y, \rho)$ was taken, associating each density level with a single depth (which varies meridionally). This allows $\psi_{iso}(y, \rho)$ to be interpolated from the meridionally varying heights $\overline{H_\rho}(y, \rho)$ to the fixed vertical grid.

For the 2D simulations, the streamfunctions $\psi_{iso}$ and $\overline{\psi}$ are calculated in exactly the same way. They are then scaled by a factor $L_x$ so that their magnitudes are comparable to the 3D streamfunctions.

Finally, the difference between the $\overline{\psi}$ and $\psi_{iso}$ is defined as the eddy-induced circulation $\psi_{eddy}$ (Abernathey et al., 2011):

$$\psi_{eddy} = \psi_{iso}(y, z) - \overline{\psi}(y, z). \tag{4}$$

An equivalent way of writing this is that the total averaged layer transport has two components. First, there is the product of the individual averages of the meridional velocity and layer thickness $\overline{v}$ and $\overline{H_\rho}$, $\overline{v}\overline{H_\rho}$. The second component is the eddy component $\overline{v'H'_\rho}$, the departure from the average. Overall we have $\overline{vH_\rho} = \overline{v}\overline{H_\rho} + \overline{v'H'_\rho}$, where the dash indicates departure from the time and zonal-average in 3D and time-average in 2D. Note that, because $\psi_{\mathrm{eddy}}$ contains a departure from the time-mean component of the total circulation in both 2D and 3D simulations, it is sometimes referred to hereon as the transient component of the circulation. In particular, here, $\psi_{\mathrm{eddy}}$ includes the drift of system away from the initial conditions. This (small) contribution, which does not represent SMLEs, is noticeable in some simulations (see below).

## 3 Results

In this section the results are discussed first for the Arctic summer experiment, starting with a description of the SMLE development and then going on to outline the eddy impact on air-sea fluxes and sea ice, before describing the role of the eddy dynamics. Secondly, the results for the Arctic winter experiment are given, in the same order. Next, Antarctic experiments are compared to Arctic ones, first summer and then winter. Finally, results of the sensitivity of the eddy impact to MLD, atmospheric forcings and initial sea ice thickness are presented.



### 3.1 Arctic summer

#### 3.1.1 Submesoscale eddy development near to the ice edge

In summer, the atmospheric forcing generates a meridional density gradient in the ML due to the presence of sea ice over the northern half of the domain. The atmospheric forcing drives ice melt and freshwater input to the surface of the ice-covered ocean while the penetrating shortwave heating warms the open ocean faster than the ice-covered ocean. Because the released freshwater is very buoyant and remains close to the surface, the developing ML meridional density gradient is salinity dominated near the surface (lighter water on the ice-covered ocean side) but is temperature-dominated (lighter water on the open

ocean side) from a few meters of depth to the bottom of the ML (Fig. (2a, 3a)). A consequence of the developing freshwater stratification close to the surface is that the MLD rapidly drops to a few metres in just a few days in both ice-free and ice-covered areas (Fig. 3c).

Through the ML baroclinic instability, ML eddies develop near the ice edge, where the meridional density gradient is the strongest. Fig. 4a displays the Rossby number for the summer simulation, clearly picking out the SMLE vortices that have

been energised through the ML instability. The Rossby number is on the border of the submesoscale range at $Ro \approx 0.3$. By day 50, eddies are well developed, as illustrated by a snapshot of SST (Fig. 4c). Eddy filaments and vortices grow at the interface of relatively warm water in the open ocean and water at freezing temperature underneath the ice cover. The vertical eddy buoyancy flux $\overline{w'b'}^{xt}$ (where prime denotes departure from the zonal mean, $w$ is vertical velocity, and $b = \frac{-g\rho}{\rho_0}$ is buoyancy) is shown at the surface and at $z = -28.75$ m in Fig. 5a. The time average is taken over the whole simulation period of 110 days.

If the vertical eddy buoyancy flux is positive (bringing light water up or dense water down), then the eddies act to restratify the ocean ML. As mentioned, the meridional density gradient is dominated by salinity near the surface and temperature below the surface. Fig. 5a illustrates that 1) eddies do not reach the full meridional extent of the domain by the end of the simulation in summer, 2) at the surface (salinity dominated) the vertical eddy buoyancy flux can be negative (i.e., destratifying), and 3) below a few meters, it is strictly positive (i.e. restratifying) with the strongest effect near the ice edge. It is noteworthy that,

despite the MLD dropping to a few meters within days, the SMLEs extend over the full initial MLD of 55 m for the entire simulation (more details in section 3.1.3). The MLD is defined as the depth at which there is a significant vertical stratification, but the ocean is weakly stratified from below the MLD down to 55 m depth, hence it remains prone to ML instability.

The tendency of eddies to release available potential energy (PE), flatten isopycnals, and restratify the ML is therefore felt through the full depth of the initial ML. This is illustrated by comparing 2D and 3D density snapshots at day 50 (Fig. 2a). In

the 2D run, the horizontal density gradient is more localised and stronger compared to that in the 3D run.

#### 3.1.2 Eddy impact on air-sea heat flux and sea ice

To evaluate the role of eddies, it is useful to explore the heat budget of the ML. We split our domain into two boxes. The open ocean region extends from the southern solid wall to the middle of the channel ($Y_b = 38$ km). The equivalent region for the ice-covered ocean extends north of $Y_b = 38$ km. The two boxes extend from the ocean surface to a depth $Z_b = 55$ m (the





**Figure 2.** Snapshots of potential density (referenced at the surface) at day 50 of the 3D zonal mean (left) and 2D (right) simulations, for (a) Arctic summer (b) and Arctic winter.

summer initial MLD). For each of these regions of volume $V$, the volume-averaged ML heat budget in units of W m$^{-2}$ is:

$$\underbrace{\rho_0 c_w Z_b \frac{\overline{\mathrm{d}T}^{xyz}}{\mathrm{d}t}}_{storage} = \underbrace{\overline{Q_{net}}^{xy}}_{surface} - \underbrace{\rho_0 c_w \overline{wT}^{xy}\big|_{z=-Z_b}}_{VHT} - \underbrace{\left(\frac{\rho_0 c_w Z_b}{Y_b}\right)\overline{vT}^{xz}\big|_{y=Y_b}}_{MHT} + F_{diff.}, \tag{5}$$



**Figure 3.** (a,c) Arctic summer, (b,d) Arctic winter. The left panel (a,b) displays the time evolution of the zonally averaged meridional density gradient in 2D and 3D simulations at $y = 38$ km, and $z = -26.25$ m (summer) and $z - 51.25$ m (winter). The right panel (c,d) displays the time evolution of ice-covered and open ocean spatially averaged MLDs for 3D (solid lines) and 2D (dashed lines) simulations.

where $\frac{\mathrm{d}T}{\mathrm{d}t}$ is the temperature tendency, $c_w$ is the heat capacity of seawater, and $Q_{net}$ is the total surface ocean heat flux. Overlines denote spatial averages. The advective terms are the vertical heat transport (VHT) and the meridional heat transport (MHT). $F_{diff.}$ are the net diffusive heat fluxes into the box. $Q_{net}$ is given by Eq. (1).





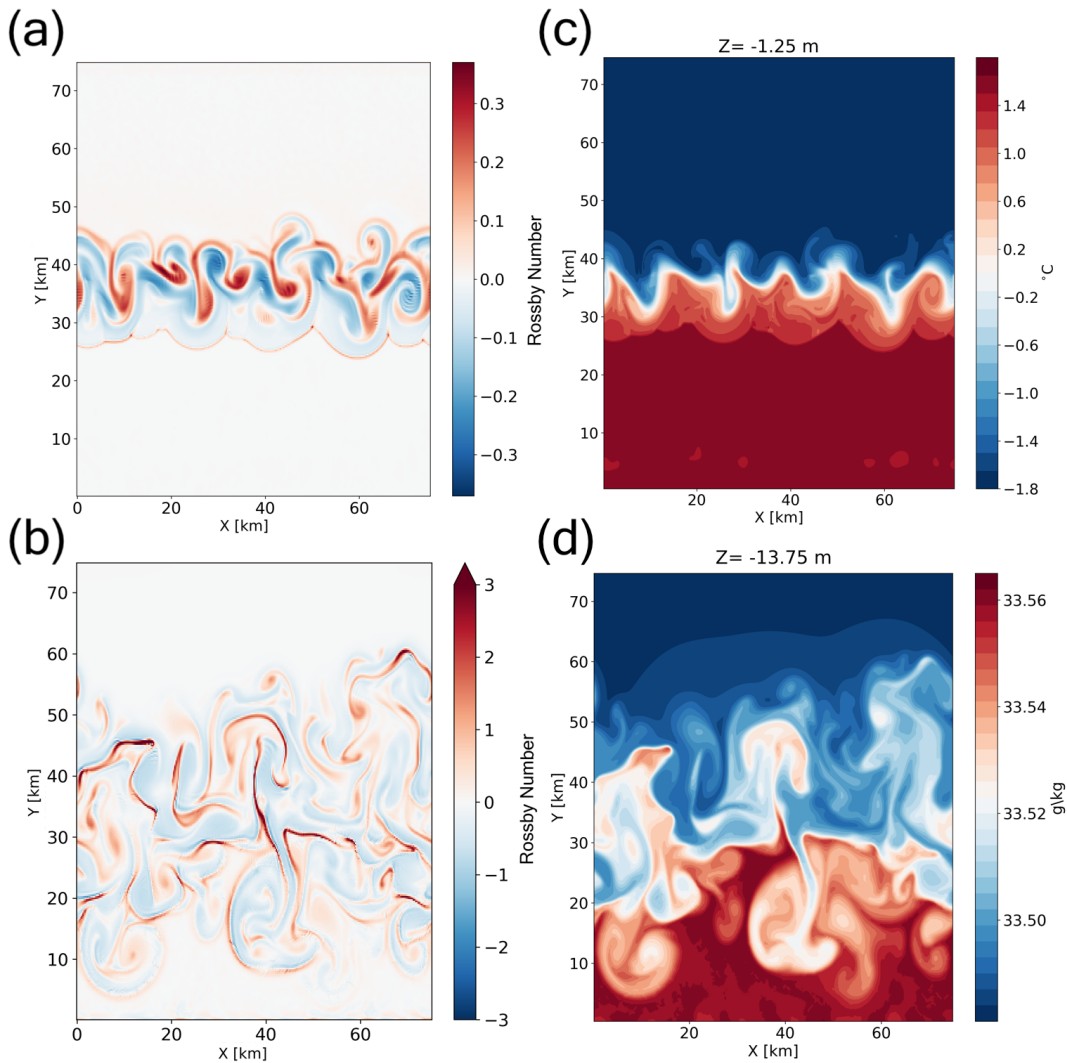

**Figure 4.** Snapshots of properties at day 50 for the Arctic case: (a,c) summer, (b,d) winter. The left panels (a,b) display the Rossby number ($=\frac{|\zeta|}{f}$) at the surface (note the differing colourbars). The top right panel (c) displays the sea surface temperature, and the bottom right panel (d) displays the salinity at a depth of $z = -13.75$ m.

Under summer conditions, $Q_{net}$ dominates the heat budget in the open ocean (Fig 6a). The dominant terms in $Q_{net}$ in the open ocean are the shortwave radiation, the sensible heat flux and the net longwave radiation. Warming due to $F_{SW,d}$ ($\mathcal{O}(10^2)$ W m$^{-2}$) dominates. However, the atmospheric temperature of approximately -10 °C brings about sensible cooling of the ocean ($\mathcal{O}(10^1)$ W m$^{-2}$), which increases in magnitude over time (from around 30 to 50 W m$^{-2}$) as the open ocean warms from (-1.8 °C to 3 °C). The increasing open ocean temperature also leads to a substantial increase in the magnitude of outgoing



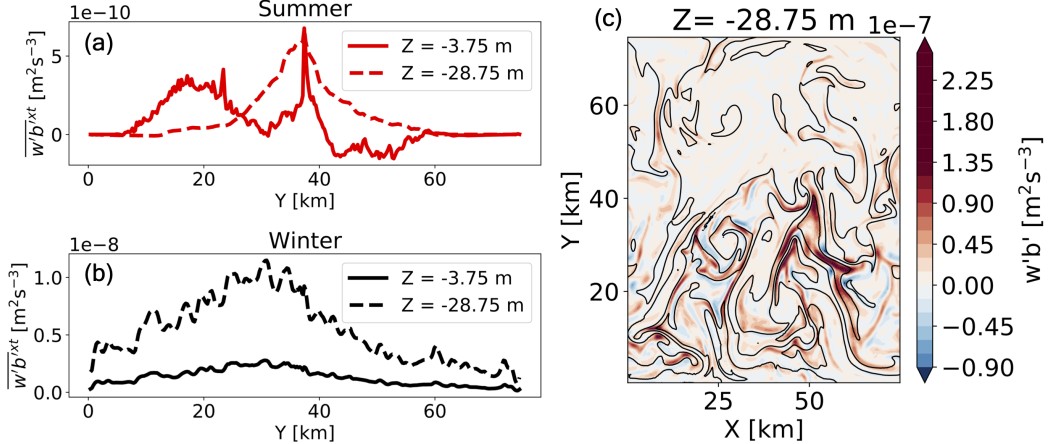

**Figure 5.** Time and zonal averages of vertical eddy buoyancy flux at the surface and at $z = -28.75$ m for (a) Arctic summer and (b) Arctic winter. The time average is taken over the full simulation length of 110 days. Note the different scales on the y-axis between a) and b). (c) spatial variations of $w'b'$ in the Arctic winter simulation at day 110, with salinity contours, spaced at 0.02 g kg$^{-1}$, shown in black.

longwave radiation (-70 to -95 W m$^{-2}$). Warming due to $F_{SW,d}$ ($\mathcal{O}(10^2)$ W m$^{-2}$) is partially conpensated by a cooling by $F_{LWnet}$ and $F_{SH}$, leading to a decrease of $Q_{net}$ over time (Fig. 6a).

The heat storage is mainly driven by $Q_{net}$, but with a significant contribution from MHT. MHT (negative, pushing heat from the open ocean to the ice covered region) reaches up to -35 W m$^{-2}$ in the 3D simulation, but only -9 W m$^{-2}$ in the 2D simulation. VHT indicates a small exchange of heat between the ML and the deeper ocean (around 1 W m$^{-2}$- not shown in

Fig. 6). Vertical mixing terms were found to be negligible through the bottom boundary $Z_b$, and the surface correction term due to the linear free surface was also found to be negligible (not shown).

Overall, the heat storage decreases by almost half over the 110 days (from 90 to 40 W m$^{-2}$) as $Q_{net}$ reduces over time and MHT strengthens.

Now we consider the dominant heat budget quantities in the ice-covered ocean region (Fig. 6b). With ice, the net surface

heat flux $Q_{net}$ is almost 2 orders of magnitude smaller than over the open ocean (only a few W m$^{-2}$) due to reflection and absorption by the ice. As the ice cover melts and thins (due mostly to in-situ heating from shortwave radiation), $F_{SW,d}$ into this region increases by an order of magnitude to $\mathcal{O}(10^1)$ W m$^{-2}$, but cooling via $F_{LWnet}$ and the sensible heat flux also increases when the ice cover melts leaving the open ocean surface. Overall the ice-covered $Q_{net}$ changes little over time. Whilst the ocean surface temperature remains at the freezing point (as the surface heating is balanced by ice melt such that the surface

temperature changes only slightly due to the salinity-dependence of the freezing point), the storage term in the ice-covered ocean is positive, dominated by the MHT from the open ocean region, and indicates overall warming of the ML especially below a few meters at the surface.





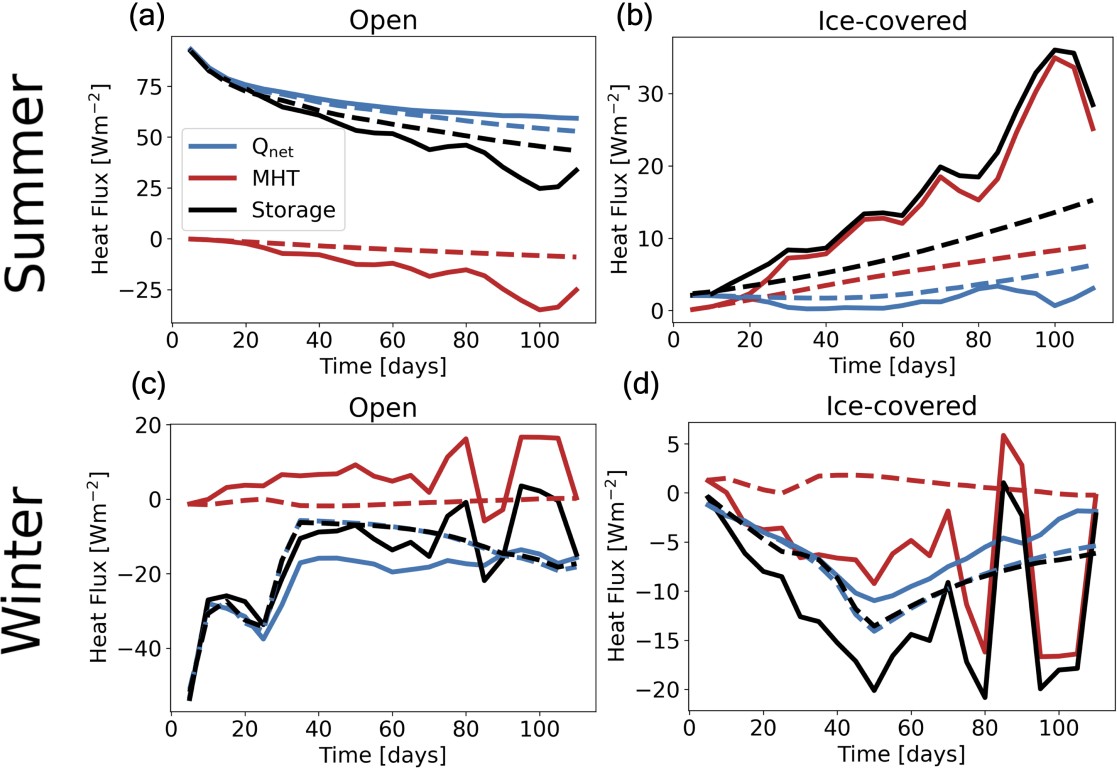

**Figure 6.** Top row: time evolution of Arctic summer open ocean (a) and ice-covered ocean (b) ML heat budgets. Bottom row: time evolution of Arctic winter open ocean (c) and ice-covered ocean (d) ML heat budgets. Solid and dashed lines denote the terms for the 3D and 2D simulations, respectively. For all quantities, 5-day averages are shown.

The impact of eddies on the heat budgets of both regions is observed by comparing the 3D and 2D results in Figs. 6(a,b). In the ice-covered region heat budget, the MHT in the 3D simulation reaches a time average of three times that of the 2D simulation by day 110. As a result, the time-averaged heat storage in the 3D ice-covered region doubles compared to 2D by day 110 (compare red solid and dashed in Fig. 6b). This demonstrates the large impact of eddies on the ice-covered region heat budget. An accelerated increase in ice melt occurs due to this eddy-induced lateral heat transport under the ice. The ice melt, in turn, impacts air-sea fluxes: a positive feedback is created between ice melt and $F_{SW,d}$ as thinner and sparser ice coverage increases the ML absorption of shortwave radiation. On average over the time of the simulation, $F_{SW,d}$ is increased by 20% in 3D compared to 2D.

The time series of total sea ice volume summed over the full domain for both 3D and 2D Arctic summer simulations is shown in Fig. 7. For 2D, we scale up by the number of zonal grid points in 3D to obtain an equivalent volume. The difference in ice volume between the 2D and 3D simulations increases over time, reaching a total difference of 10% by day 110. However, this value of 10% is impacted by the region included in its calculation. Fig. 8 displays the sea ice volume at day 110 in 3D



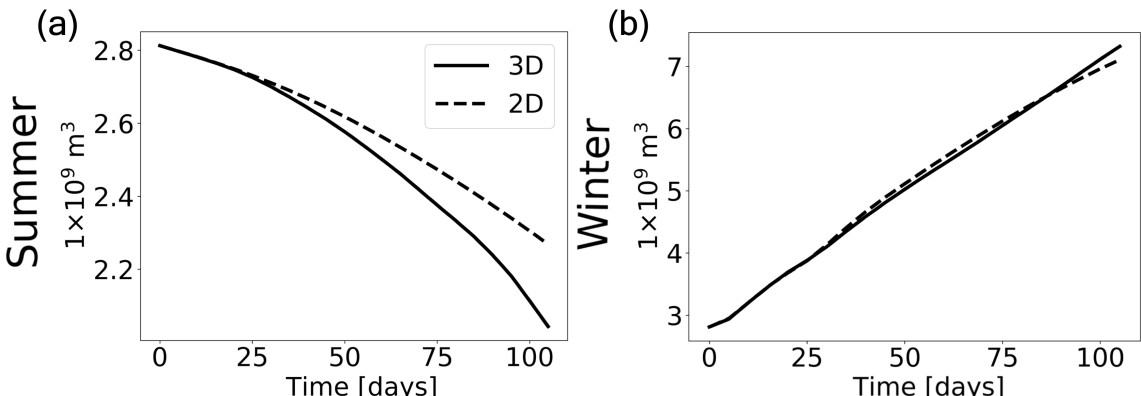

**Figure 7.** Total 3D (solid) and 2D (dashed) sea ice volume time series for (a) Arctic summer and (b) Arctic winter.

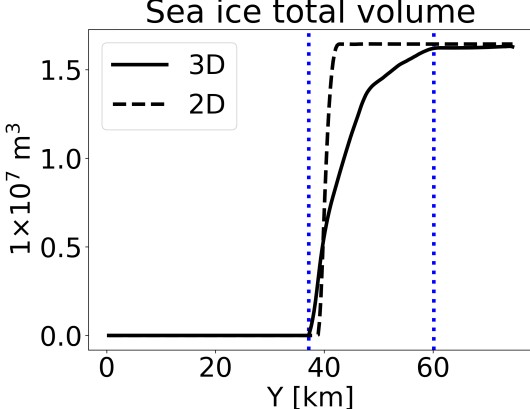

**Figure 8.** Day 110 snapshot of the zonal sum of sea ice volume, 2D and 3D Arctic summer (black lines). Blue dashed lines are an illustration of the meridional extent of the eddies.

and 2D Arctic summer summed in the zonal direction. The meridional extent of the eddies underneath the ice cover is shown by the blue dashed lines. When only considering the region bordered by these lines, the final 2D-3D sea ice volume difference rises to 17% (below we also consider the influence of the choice of initial ice thickness on this number). Nonetheless, Fig. 8 emphasizes that the impact of SMLEs on the sea ice margin is tightly linked to their spatial scale and growth rate.

### 3.1.3 Eddy dynamics

We now look more closely at the role of the SMLEs. In Fig. 9a, the clockwise direction of $\psi_{eddy}$ indicates eddy-induced circulation acting to raise less dense (temperature-dominated density) water on the open ocean side and to sink denser water under the ice. As expected, the eddy-induced circulation acts to flatten the density front and feeds on the source of baroclinic instability, the available PE. $\psi_{eddy}$ extends over the bulk of the ML below the ∼5 m thick freshwater lens and is centred





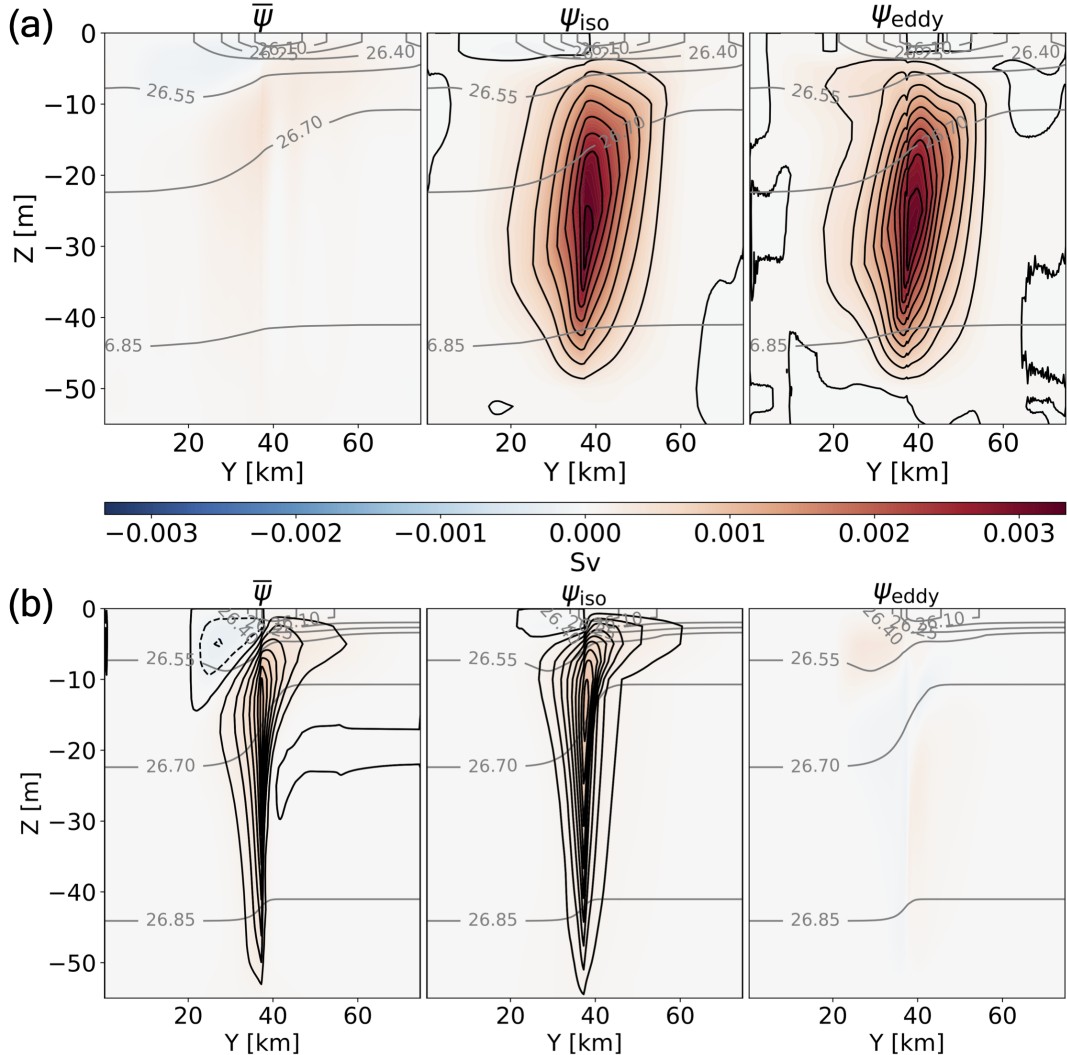

**Figure 9.** The Arctic summer time-averaged overturning stream functions for (a) 3D and (b) 2D simulations. $\overline{\psi}(y, z)$ (left panels), $\psi_{\mathrm{iso}}$ (centre panels), and $\psi_{\mathrm{iso}}$ (right panels). Red/blue shadings indicate clockwise/counterclockwise cells. Time-averaged isopycnals are shown in grey. The time period of averaging is the full simulation length of 110 days.

underneath the initial ice edge. The total overturning, $\psi_{iso}$, is dominated by $\psi_{eddy}$, and $\overline{\psi}$ is negligible in comparison. Fig. 9a

demonstrates that the oceanic circulation, and associated heat transport to the ice-covered ML are largely due to the effect of eddies.

Fig. 9b (2D Arctic summer simulation) shows that the Eulerian overturning stream function associated with the density front is almost twice as large than in the 3D simulation, scaling with the stronger lateral density gradient in the 2D simulation. Furthermore, the total overturning $\psi_{iso}$ in the 2D simulation is approximately three times smaller than $\psi_{iso}$ in the 3D simulation.



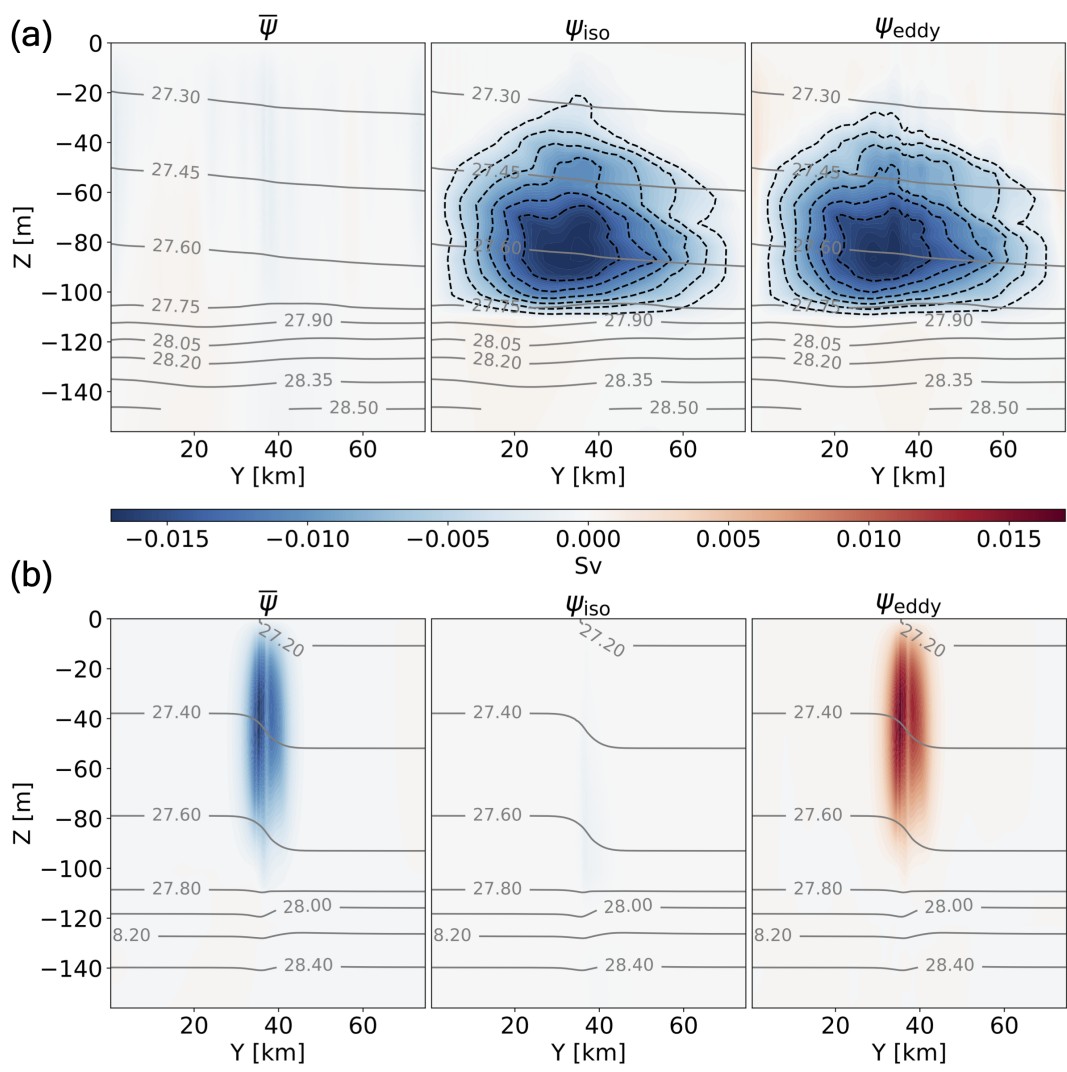

**Figure 10.** As in Fig. 9 but for Arctic winter. Note the different colorbar range compared to Fig. 9.

In 2D, there are no eddies, and $\overline{\psi}$ is approximately equal to $\psi_{iso}$. Note that $\psi_{eddy}$ in 2D is negligible, but is not exactly zero, because of the transient component of the layer transport (see section 2.1). Nonetheless, this difference in $\psi_{iso}$ between the 3D and 2D simulations corresponds well to the difference in magnitudes of the meridional heat transport between simulations displayed in the heat budget.



## 3.2 Arctic winter

### 3.2.1 Submesoscale eddy development near the ice edge

Like in summer, the winter initial conditions are horizontally uniform in temperature and salinity. Under winter atmospheric forcings, the ocean cools and ice forms within a few days. The rate of ice formation, which is associated with the rate of brine rejection, is faster in the open ocean, where ice is not already present or is thin. This is related to the conduction flux through ice which is inversely proportional to ice height. The salinity of the open ocean, therefore, increases faster than that of the ice-covered ocean (Fig. 4d). The developing winter ML density gradient (Fig. 3b) between open and ice-covered regions is brought about by colder, saltier waters in the open ocean and warmer, fresher water in the ice-covered region (note that the density gradient is reversed compared to the summer case). Also, the gradient is salinity dominated as the lateral temperature gradient is very small (less than 0.05 °C). This is because the ML is close to freezing point in both regions. As a consequence of the unstable ML, dense water is vertically mixed, the ML deepens and warmer water is entrained from below the ML (Fig 3d).

SMLEs develop through ML baroclinic instability near the sea ice edge as in the summer. The eddies spread heat and salinity anomalies laterally (Fig. 4d), extracting the available PE from the lateral density gradient. Fig. 2b displays day 50 snapshots of potential density in 3D and 2D. Eddies develop and spread more rapidly, as well as reaching higher Rossby numbers, in the winter simulation compared to the summer simulation (Fig. 4b). As expected, the front is less steep in 3D than in 2D (Fig. 2, lower panels), which is reflected in the evolution of density gradient in the two simulations (Fig. 3b).

Fig. 5b shows the Arctic winter time zonal-mean vertical eddy buoyancy flux $\overline{w'b'}^{xt}$ at the surface and at $z = -28.75$ m. $\overline{w'b'}^{xt}$ strengthens by an order of magnitude at depth compared to at the surface, and at both depths, the restratifying flux is at least an order of magnitude greater than in summer ($10^{-8}$ m$^2$s$^{-3}$ at depth and $10^{-9}$ m$^2$s$^{-3}$ at the surface). This restratification (positive flux) by eddies opposes the destratifying influence of cooling and ice formation. In Fig. 5c, the winter spatial variations $w'b'$ are shown at day 110, with salinity contours shown in black. The alignment of $w'b'$ and salinity demonstrates that buoyancy anomalies are salinity dominated. $w'b'$ reaches up to $10^{-7}$ m$^2$ s$^{-3}$ in parts of the open ocean, where air-sea fluxes are also larger. These magnitudes of vertical eddy buoyancy fluxes are equivalent to vertical heat fluxes (W m$^{-2}$) in the submesoscale range (Fox-Kemper et al., 2011; Fox-Kemper and Ferrari, 2008; Thomas et al., 2008).

### 3.2.2 Eddy impact on air-sea heat flux and sea ice

For the winter case, we set the bottom boundary of the domains, $Z_b$, at the approximate final MLD of 147 m. This is to ensure that all the SMLE dynamics are included within the heat budget.

The rapid onset of ice cover in the open ocean has a large influence on the time evolution of the winter heat budget (Fig. 6c). All winter heat budget terms other than the net surface heat flux, the MHT and the heat storage, are negligible and are not displayed in Fig. 6. In the open ocean, the initial net air-sea flux is -77 W m$^{-2}$. It stays at this value throughout the simulation at locations that remain ice-free (because it is controlled by the SST, which remains at freezing point). At locations where ice grows, the surface heat flux decreases to approximately -10 W m$^{-2}$. This explains the initial rapid reduction of the net surface heat flux averaged over the initially open ocean (see in Fig. 6c, from over -50 W m$^{-2}$ to around -30 W m$^{-2}$). After around





40 days, $Q_{net}$ is dominated by the presence of sea ice and changes little over the remaining 70 days of the simulation. The ice growth significantly affect all components of $Q_{net}$: $F_{LWnet}$ (upwards) decreases from -120 to -82 W m$^{-2}$ (with a small effect due to decreasing SST), $F_{SH}$ decreases from -7 to -3 W m$^{-2}$ and $F_{SW,d}$ (downward) decreases by just over one-half in 2D and just under half in 3D (it ranges from 50 W m$^{-2}$ into the open ocean to less than 10 W m$^{-2}$ into ice-covered ocean). As in the summer case, $Q_{net}$ is the dominant term in the heat budget in the open ocean region although with weaker values ($\mathcal{O}(10^1)$ W m$^{-2}$).

The ML open ocean heat storage term similarly remains large over time ($\mathcal{O}(10^1)$ W m$^{-2}$). This implies fast cooling of ocean waters below the base of the initial MLD (which are included in the heat budget volume average) through vertical mixing as the ML deepens. The temperature tendency near the ocean surface only is actually close to zero, because of the balance between ice formation, heat lost at the surface, and heat entrained from below the initial MLD. As mentioned above, the depth-averaged lateral temperature gradient is weak (and reversed) compared to the summer case at about -0.05 °C (+3 °C in summer). As a result, the MHT acts to warm the open ocean, but it is smaller magnitude in winter than in summer.

In the winter ice-covered ocean region (Fig. 6d), the ice thickens over time. Heat is conducted from the ocean through the ice and is lost to the atmosphere. Rates of ocean cooling (heat storage term) are lower than in the open ocean (by around 20 W m$^{-2}$) due to a complete and thicker ice cover. The heat storage term is dominated by the net surface heat flux $Q_{net}$ and MHT in 3D (both cooling the ice-covered box), and only by $Q_{net}$ in 2D. Again, the volume-averaged heat storage term also encompasses heat mixed upwards and lost at the surface, through the process of deepening the ML. As in the winter open ocean region, the heat storage term nearer to the surface is much smaller due to the balance between heat loss through sea ice and entrainment of warmer water from below.

The MHT at the initial ice edge ($Y_b$ =38 km) is small in 2D, less than 1 W m$^{-2}$, but up to 17 W m$^{-2}$ in 3D. This southward eddy-driven MHT acts to partially cancel the heat loss to the atmosphere that drives ML deepening in the open ocean, resulting in arresting MLD deepening there (see Fig. 3d). The eddy-driven cooling of the ice-covered ocean has a significant impact on the heat budget, reinforcing the heat loss through ice. The ML depth average heat storage in the ice-covered region decreases further, by around 30% on average over time, in 3D compared to 2D (Fig. 6d).

The overall effect of the eddies on sea ice growth is however small. Eddies increase the total ice volume (over the whole domain) by only 3% by day 110. This is due to compensating effects. As more heat is transported laterally into the open ocean from under the ice in 3D than in 2D, less heat is obtained from below the ML in 3D, because eddies restratify and the ML does not deepen as much. Nearer to the ice edge, the balance between these lateral and vertical heat exchange terms is very fine, but further south the difference is more stark and the differences between the 2D and 3D sea ice volume are larger. In the ice-covered region, rates of surface heat loss and also basal heating are a smaller factor in the heat budget, so lateral eddy heat transport is more important for heat storage and ice formation rates. This means there is also higher ice growth and rates of heat loss in 3D in the winter ice-covered region.




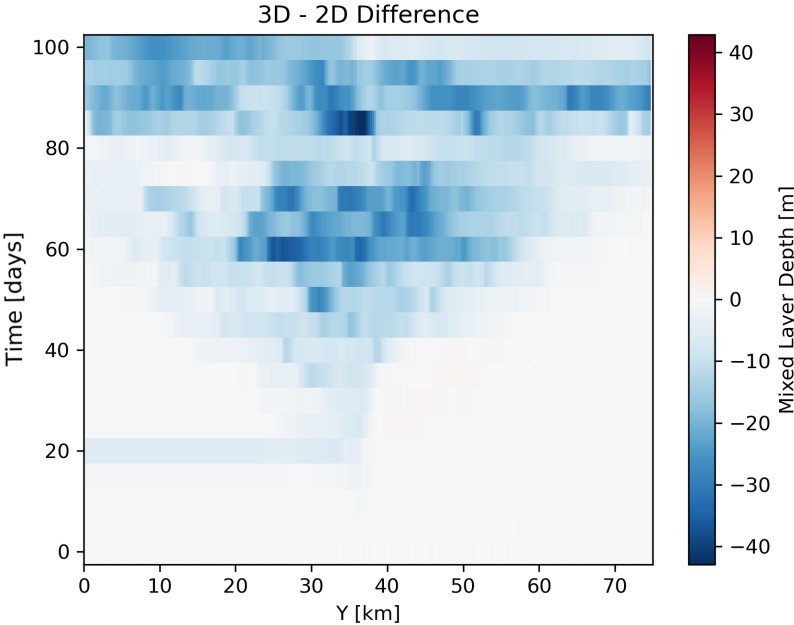

**Figure 11.** Time evolution of difference in MLD (zonally averaged) between simulations with eddies (3D) and no eddies (2D) for the Arctic winter case.

### 3.2.3   Eddy dynamics

The 3D and 2D Arctic winter overturning stream functions are displayed in Fig. 10(a,b), respectively. The 3D Eulerian stream-function (Fig. 10a, left) is 2 orders of magnitude less than $\psi_{iso}$ and $\psi_{eddy}$, both of which are of order $\mathcal{O}(10^{-2})$ Sv. As in the summer simulation, the winter net overturning is dominated by the eddy dynamics in 3D. The direction of the winter eddy-induced overturning is anticlockwise (opposite direction to the summer simulations, reflecting the reversed winter lateral density gradient). In the winter simulation, eddies act to drive fresher, warmer water upwards in the ice-covered region and

cooler, saltier waters downwards in the open ocean region.

It is worth reiterating that, in 2D, where SMLEs are absent, only time changes contribute to the eddy-induced overturning. Fig. 10b shows that whilst in summer this transient contribution was negligible ($\overline{\psi}$ is approximately equivalent to $\psi_{iso}$ in 2D), in winter the transient contribution is important. In particular, when averaged over the entire simulation, the transient contribution is large (but, again, by construction, it is zero when calculated instantaneously). The '$\psi_{eddy}$' in 2D is positive with a narrow

meridional extend. This transient component has a weakening impact on the 3D eddy-induced overturning (shown in Fig. 10a, right), which would be stronger instantaneously. Overall the winter stream functions provide a helpful characterisation of the 3D eddy dynamics and confirm that, as in the summer simulation, the total overturning in 2D (no eddies) is an order of magnitude weaker than in 3D.





Fig. 11 shows the time evolution of the difference in MLD between the 2D and 3D simulations (blue shading indicates
deeper MLDs in 2D). The meridional expansion of the region with deeper 2D MLDs reflects eddies travelling further away
from the initial ice edge over time. Eddies reduce the deepening of the ML by 80% when averaged over the open ocean region
and 52% when averaged over the ice-covered region by day 110. The difference between regions is because cooling and ML
deepening are more rapid in the 2D open ocean region than in the ice-covered one. In both regions, the final 3D MLD is only
5 m deeper than the initial one, implying that eddies nearly balance the destratifying impact of atmospheric cooling and ice
formation in both regions.

## 3.3 Antarctic

This section outlines the results for Antarctic summer and winter simulations, emphasizing how they differ from their Arctic
counterparts. Initial temperature and salinity profiles used in the Antarctic simulations are displayed in Fig. 1.

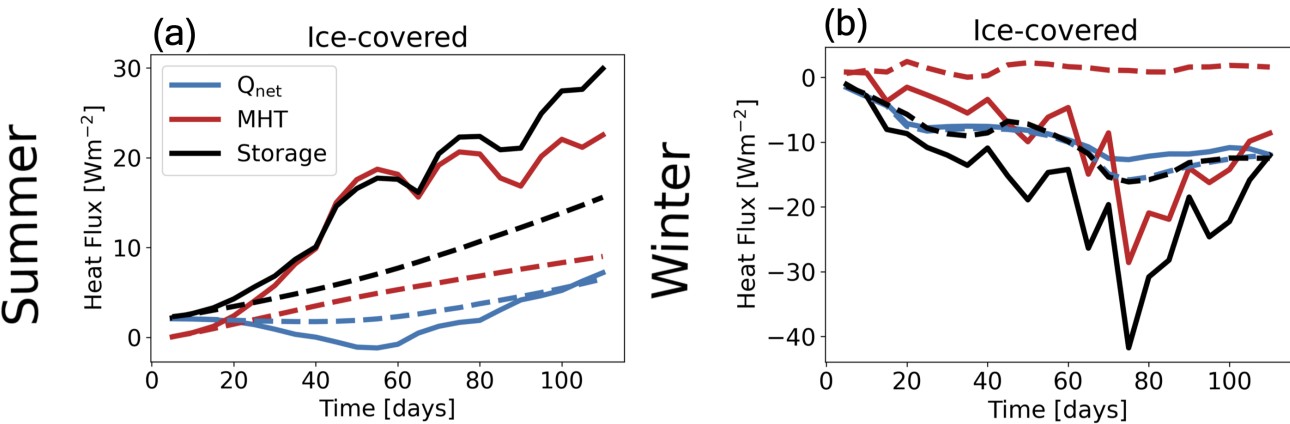

**Figure 12.** Ice-covered ML heat budgets for (a) Antarctic summer and (b) Antarctic winter (as in Fig. 6 but for the Antarctic simulations).
Solid and dashed lines denote the 3D and 2D results, respectively.

### 3.3.1 Summer

Under summer conditions, the Antarctic initial stratification produces only small differences compared with the Arctic case.
For example, eddies decrease the final ice volume by 11% with Antarctic initial stratification, compared to 10% in the Arctic
case. The MHT to the ice-covered region is only 4% larger for the Antarctic than the Arctic 3D simulation on average. The
time-averaged open ocean heat storage, ice-covered ocean heat storage, and $\psi_{iso}$ exhibit negligible differences between 3D
Arctic and Antarctic simulations too. Therefore, eddies also have a significant impact on the Antarctic summer ice-covered
region heat budget, as illustrated in Fig. 12.





This low sensitivity of the eddy impacts to initial conditions in summertime is due to the tendency of the summer atmospheric forcings to enhance ML stratification in the ML and reduce the MLD abruptly. This implies that there is little interaction between the ML and the subsurface ocean in the summer simulations. It is important to note that these findings may be limited by the simplifications of our model set-up. The simulations do not take into account the wind forcing, nor the difference in wind forcing between locations. A wind forcing would contribute to sustain the MLD against the restratifying effect of the surface buoyancy forcing, effectively reducing the strong decoupling we observe between the ML and the subsurface.

### 3.3.2 Winter

In Antarctic winter, as the ML deepens, heat from the subsurface ocean is mixed upwards. The rate of vertical mixing depends on the strength of the stratification, which is weaker in the Antarctic initial profiles than in the Arctic ones. Fig. 13d displays the rate of ML deepening in the Antarctic simulations, which is significantly larger than in the Arctic simulations (compare with Fig. 3). In the Antarctic simulations, the final open ocean MLDs are approximately 160 m in 2D and 149 m in 3D. In the Arctic simulations, they are 118 m in 2D and 103 m in 3D (Fig. 3d). As in the Arctic simulations, the 2D-3D difference in MLD, a proxy for eddy restratification, is greater in the open ocean region than in the ice-covered region. Note that while eddies reverse the initial MLD deepening in the Arctic winter simulation, this does not happen in the Antarctic case because of the faster ML deepening, even if the total ML shallowing effect by eddies is of the same magnitude for both locations (15 m for the Arctic and 11 m for the Antarctic simulations).

The greater vertical mixing of heat upwards from the subsurface ocean in the Antarctic case results in a slower ice formation, and hence in weaker 3D and 2D lateral density gradients. This effect can also be seen on the storage term in Fig. 12b, which is of greater magnitude in the Antarctic than in the Arctic simulations. That said, it is worth to note again that eddies have a small impact on sea ice cover in the winter simulations: by day 110, the 2D-3D difference in sea ice formation is -3% in the Arctic, whilst it is +3% in the Antarctic.

The Antarctic eddy-induced overturning stream function is overall deeper (stronger MLD deepening) and (slightly) weaker than the Arctic one (Fig. 13c).

### 3.4 Sensitivity to MLD, atmospheric forcings and initial sea ice thickness

Finally, in this section, the sensitivity of the Arctic summer and winter experiments to different initial values of atmospheric forcing and MLD is explored. More specifically, we test, for Arctic summer case, the sensitivity of the eddy impact on ice melt to different values of $F_{SW,d}$ (180 to 270 W m$^{-2}$), of the initial MLD (10 to 100 m), and of the initial sea ice thickness (0.5 and 1 m). For Arctic winter, the sensitivity of the eddy impact is investigated for initial MLD values of 55 to 155 m.

Fig. 14 summarizes the results of the sensitivity experiments. Fig. 14(a,b) display the Arctic summer 2D-3D percentage difference of final total sea ice volume for different values of $F_{SW,d}$ and different initial values of MLD. There is a near monotonic increase of the eddy impact on sea ice for increasing $F_{SW,d}$ or initial MLD. The 2D-3D difference reaches 30%





**Figure 13.** Results for the Antarctic winter case. (a) time evolution of total sea ice volume for both 3D (solid line) and 2D (dashed line) simulations (compare with the Arctic case in Fig. 7b). (b) time evolution of the zonally averaged meridional density gradient at $y$=38 km, and at $z$=-51 m (compare with Fig. 3(b)). (c) the 3D eddy-induced overturning (compare with Fig. 10(a), right panel). (d) time series of MLDs for the open and ice-covered ocean (compare with Fig. 3(d)).

for $F_{SW,d} = 270$ W m$^{-2}$ (due to stronger lateral density gradients and eddy MHT), and 14% for MLD=100 m (again, due to
stronger eddy overturning and associated MHT for deeper initial MLD).

Further, when the initial ice thickness in the Arctic summer simulation is decreased from 1 m to 0.5 m, the 2D-3D percentage ice volume difference grows to 60%. This demonstrates a strong dependence of the percentage impact of eddies on the initial ice thickness. Eddies actually melt about half as much ice overall in volume when the ice thickness is halved (the 2D-3D final ice volume difference is about half when initial ice volume is halved). However, melt due to non-eddy effects (e.g., positive
shortwave radiation feedback) is much greater over time with a thinner ice cover. So the final ice volume in 2D with 0.5 m





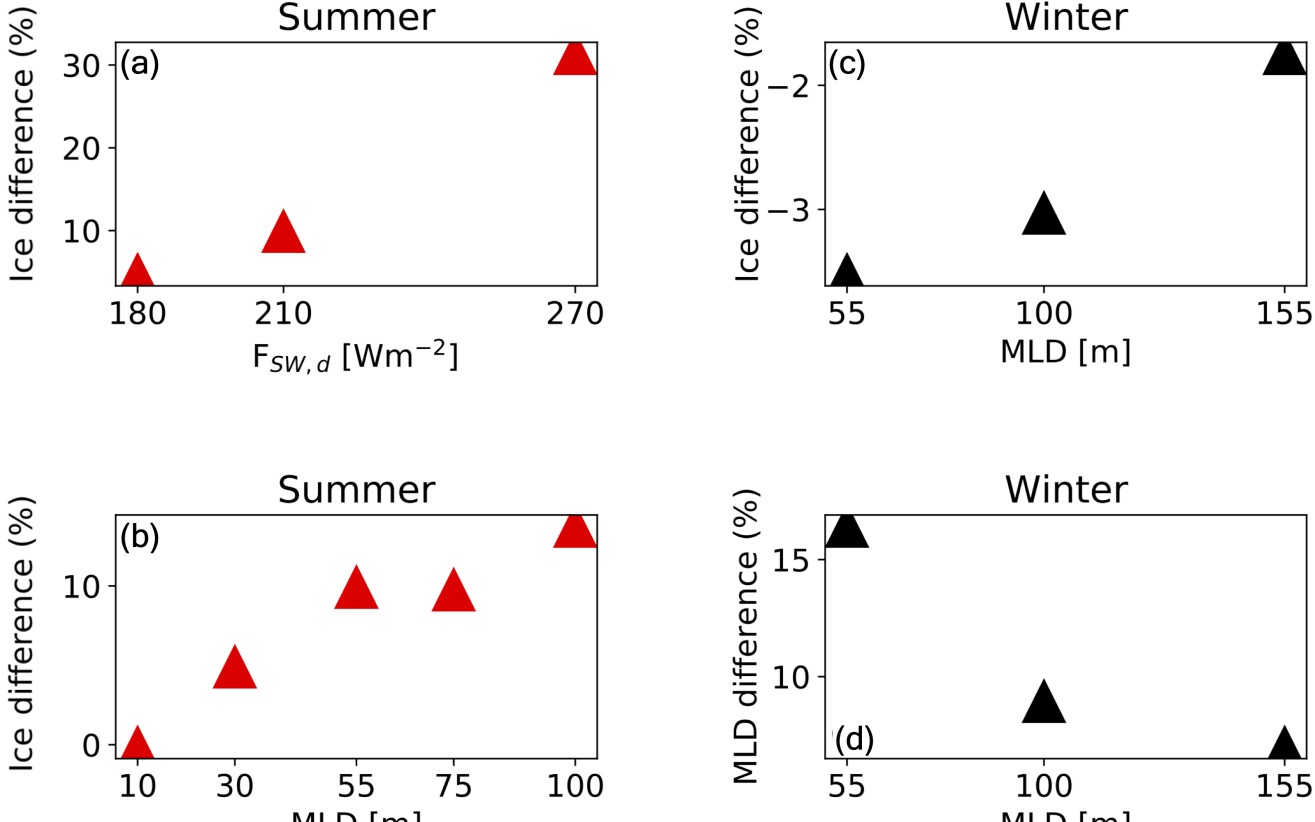

**Figure 14.** Sensitivity of Arctic summer 2D-3D ice volume difference to (a) $F_{SW,d}$ and (b) initial MLD. Sensitivity of Arctic winter 2D-3D ice volume difference (c) and 2D-3D difference in the final MLD (d), to the initial MLD. All quantities are spatially averaged.

initial ice height is less than half the final ice volume in 2D with 1 m initial ice height. This means the eddy melt as a fraction of the 2D final ice cover increases with 0.5 m initial ice height (even if in absolute volume the eddy melt is around half).

For the Arctic winter case, Fig. 14c displays the sensitivity of the eddy impact on ice formation to the initial MLD. Unlike the summer case, the impact of eddies on the total amount of ice formed is not sensitive to the initial MLD. The percentage

ice difference varies between 1.7-3.5% overall, effectively a doubling but this should be compared to a change from 0 to 15% in the summer case (panel b). Fig. 14d shows how the percentage difference in MLD due to eddies decreases as initial MLD is increased - the figure ranges from 16% with an initial MLD of 55 m to 7% with an initial MLD of 155 m. This reflects how the eddies do not proportionally restratify the ML by the same fraction with increasing MLD. Instead, the absolute reduction to MLD by eddies is almost constant for all tested initial MLDs, 10±1.5 m.




## 4 Summary and discussion

In this study, the impact of submesoscale mixed layer eddies (SMLEs) generated near the sea ice edge on the ocean, sea ice and air-sea exchanges is explored using idealized numerical simulations. Our focus is on the thermodynamical coupling between SMLEs and sea ice, which has received little attention in previous studies compared to mechanical coupling (hence we do not consider the wind forcing).

We use submesocale-resolving simulations (at 250 m resolution) of the ocean mixed layer (ML) near the ice edge, representing a lead or the marginal ice zone. At the initial state, the northern half of the domain is covered with ice. 3D simulations with eddies are compared to 2D (no zonal variation, but otherwise identical) simulations where eddies are absent.

Four main experiments were used in order to understand how SMLEs behave near the sea ice edge: Arctic and Antarctic summer and winter. Between locations, only the initial stratification differs, whilst, between seasons, the initial stratification, atmospheric forcings and initial mixed layer depths (MLDs) differ. The season affects the sources of heat available to or accessed by the ML (i.e., atmosphere and subsurface ocean). Further sensitivity tests helped clarify the influence of these ambient conditions on our results. The key outcomes of our study are:

- The Arctic summer simulation shows that SMLEs energised near the ice edge have a leading order-to-moderate impact on air-sea heat exchange, ocean heat storage and lateral heat transport. In summer, although the ML shoals over time, the initial MLD (55 m) sets the vertical extent of the SMLEs and hence of the eddy-induced overturning, and this extent does not change over the simulation. The eddies have little influence on MLD but still act to increase stratification over the full 55 m depth near the ice edge and under the ice, as well as restratifying near the surface through eddy transport of freshwater to the open ocean.

- Initially, open ocean storage rates are close to order $10^2$ W m$^{-2}$, nearly 2 orders of magnitude greater than in the ice-covered region. This heat reservoir is efficiently made available to the ice-covered ocean by eddies; the rate of northward heat transfer is 3 times larger with than without eddies. When eddies are present (3D versus 2D), the net surface heat flux in the open ocean is 12% greater and the shortwave absorption in the ice-covered region increases by 20%.

- Effectively, eddies act as a heat pump: taking heat out of the atmosphere to move it under the sea ice, which in turn accelerates the melting of sea ice. This creates a positive feedback with an increase in shortwave absorption over the thinner ice. Comparing the 3D and 2D simulations shows that, in about 3 months, eddies contribute a further decrease of the ice volume of 17% in the region where they are present. This effect is strongly sensitive to the solar forcing and thickness of the initial ice cover. The percentage triples when the solar forcing increases from 180 W m$^{-2}$ to 270 W m$^{-2}$ or increases even more when the initial ice cover decreases from 1 to 0.5 m.

- In summer conditions, the Antarctic case shows a very similar behavior to the Arctic case. The background stratification (i.e., location) has little influence on the simulations because the ML shallows.

- In winter, we find a radically different situation. The heat lost to the atmosphere over the open ocean is partially balanced by heat stored below the ML (as the ML deepens) and by heat transported by eddies from the ice-covered region. Because



of the restratifying effect of eddies, less heat is transferred upwards into the ML from the subsurface ocean in 3D than in 2D. In the Arctic winter simulations, eddies have a leading order impact on MLD, reducing the ML deepening by 80% or approximately 15 m. In the Antarctic, the reduction of deepening is similar to that of the Arctic ($\sim 11$ m) but the percentage difference in final MLDs is smaller ($\sim 18\%$) due to larger initial MLDs.

– In winter, the direction of eddy-induced overturning is opposite and at least an order of magnitude larger than in summer, with a faster pace of eddy growth and subsequent transport of density anomalies. This is consistent with an eddy growth rate proportional to the MLD, which is larger in winter. However, the eddy-induced MHT is weaker than in summer because of the smaller meridional temperature gradient between regions (both at the freezing point). The eddy-induced meridional heat transport is nonetheless significant as the impact of eddies on air-sea fluxes in winter is of similar magnitude to that in summer.

– There are therefore two competing eddy effects in winter: with eddies, less heat is transported from below the ML, but more heat is transported laterally from the ice-covered ocean. The former effect is marginally greater than the latter and so the net effect of eddies is to slightly increase ice growth in the open ocean region.

SMLEs are important for polar regions as well as globally due to the way they regulate heat exchange between the atmosphere and the deeper ocean (Thompson et al., 2016). Including these eddies in climate models can have significant impacts on sea ice cover, global heat budgets, air-sea exchange, and nutrient transport (Su et al., 2018; Fox-Kemper et al., 2011; Mahadevan, 2016). In addition, capturing SMLEs can energise mesoscale eddies at the larger scale and global ocean currents such as the AMOC (Lévy et al., 2012; Fox-Kemper et al., 2011).

Including eddies in climate models relies on effective parameterisations capturing key behaviours. This work helps to understand the impact of SMLEs in polar environments, and will aid with the development of SMLE parameterisations suitable for polar regions.

Specifically, an intriguing finding of our simulations is that the vertical scale of the eddies and the eddy-induced circulation is primarily set by the initial MLD. Even in summer, where the MLD drops to a few meters in days, the eddies keep the memory of the initial MLD ($\sim 50$ m) while in winter the eddy-induced overturning extends slightly over time to match the final MLD. This suggests a significant decoupling from or lag ($\sim$month) of the eddy-induced overturning behind the MLD in summer. The Fox-Kemper SMLE parameterization (Fox-Kemper et al., 2008; Fox-Kemper and Ferrari, 2008; Fox-Kemper et al., 2011; Calvert et al., 2020) is currently the most used SMLE parametrization in climate models. Although this parameterization was not developed in the context of polar regions, it is regularly used in global set-ups, hence the need to evaluate its performance near the ice edge. On-going work explores the capabilities of the Fox-Kemper SMLE parameterization to capture the behaviour we have uncovered (to be published).

*Code availability.* The MITgcm code and inputs needed to reproduce our results are provided in (Greig and Ferreira, 2025) (Note to reviewers: this citation contains a temporary URL, the dataset will be published with a permanent DOI once the review process is complete).





# Appendix A

## A1 Initial conditions

The horizontal eddy viscosity is set in both winter simulations to $1 \text{ m}^2\text{s}^{-1}$ on the divergent component of the flow and to $50 \text{ m}^2\text{s}^{-1}$ on the vorticity component of the flow. In summer Smagorinsky viscosity scheme is used, with coefficient set to 2. In addition, a small horizontal diffusivity of $1 \text{ m}^2\text{s}^{-1}$ was used in winter simulations.

*Author contributions.* LG performed the analysis and led the writing of the manuscript at the University of Reading, UK. DF supervised, proposed and guided the project, and contributed to the writing and analysis.

*Competing interests.* The contact author has declared that none of the authors has any competing interests

*Acknowledgements.* LG was supported by the Centre for Doctoral Training in Mathematics of Planet Earth, with funding from the UK Engineering and Physical Sciences Research Council (EPSRC) (grant EP/L016613/1). We thank the authors of (Horvat et al., 2016) for their code, which provided the basis for developing our set-up.



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
