# Peer review of "Seasonal impact of submesoscale eddies on the ocean heat budget near the sea ice edge"

_EGUsphere, 2025_

## Referee Comment (RC1)

**Referee Comment on "Seasonal impact of submesoscale eddies on the ocean heat budget near the sea ice edge"**

**General Comments**

This manuscript presents an interesting and timely investigation of the role of submesoscale mixed-layer eddies (SMLEs) in modulating the seasonal ocean heat budget and sea ice evolution near the marginal ice zone (MIZ). By comparing eddy-resolving (3D) and non-eddy (2D) MITgcm simulations, the authors isolate the thermodynamical impacts of SMLEs, which are often neglected in coarse-resolution climate models. The study addresses a significant knowledge gap by quantifying how SMLEs influence sea ice melting and freezing processes through heat transport and feedback mechanisms.

The paper is generally well-motivated, clearly written, and scientifically sound. The approach is methodologically appropriate, and the results provide new insights into the coupling between submesoscale dynamics and polar climate processes. The identification of distinct summer and winter responses strengthens the contribution and highlights the necessity of submesoscale parameterization in climate models.

Overall, this is a contribution of interest to the polar oceanography and climate modeling communities. Some clarifications and improvements would further strengthen the manuscript before publication.

**Specific Comments**

**[1] Introduction**

The introduction provides a comprehensive overview of submesoscale dynamics and their relevance in polar regions. However, it is somewhat verbose and would benefit from streamlining to emphasize the central research question. The authors may consider shortening the general background on SMLEs, improving transitions between topics, and highlighting the novelty of their thermodynamic focus relative to previous studies.

While observational and modeling studies are well cited, the flow from general submesoscale theory to polar-specific impacts could be smoother. Adding brief linking sentences to connect classical SMLE mechanisms with polar sea ice interactions would enhance readability. Moreover, the main research gap, which involves quantifying the thermodynamic impacts of SMLEs across seasons and background stratifications, should be emphasized earlier, ideally before detailing the study objectives.

**[2] Materials and Methods**

The description of the model setup contains many technical details in long, complex sentences. For example, information on vertical and horizontal grid spacing, mixing schemes, and viscosity settings could be split into shorter sentences or a table. This would improve readability and make it easier for readers to understand the experimental design.

Some parameter choices, such as the Smagorinsky coefficient, horizontal eddy viscosity, and small horizontal diffusivity in winter, are described, but the rationale is brief. It would strengthen the manuscript to explain why these values were selected, particularly how they affect numerical stability and the development of SMLEs, and whether sensitivity tests were conducted.

While the 2D "no eddies" configuration is introduced, the description could clarify explicitly which processes are suppressed (e.g., lateral variations, baroclinic instabilities) and ensure the naming of the experiments is consistent (Arctic/Antarctic, summer/winter). Providing a concise summary table of the main experiments with initial conditions and key parameters would greatly enhance reproducibility.

The methods section contains a large amount of technical detail, including grid spacing, viscosity and diffusivity settings, and atmospheric boundary treatments. While these details are important for reproducibility, some of the more intricate numerical specifications could be moved to the Appendix. This would streamline the main text, improve readability, and allow readers to focus on the key experimental design and scientific rationale, while still providing full information for replication.

**[3] Results**

The Results section 3.1.1 provides a detailed description of SMLE development and their impact on the mixed layer during Arctic summer, including vertical stratification and buoyancy fluxes. While the simulations and analyses appear comprehensive, the presentation could be strengthened by improving the logical flow and emphasizing the physical interpretation. Currently, the text mixes descriptions of forcing, stratification, eddy development, and vertical fluxes in a single narrative, which can make it challenging for readers to follow the causal chain. Reorganizing the section to first describe the atmospheric and oceanic forcing, then the resulting stratification and MLD evolution, and finally the eddy dynamics and their restratifying or destratifying effects would improve clarity.

The analysis of the ML heat budget and eddy impacts (Section 3.1.2) is thorough and provides valuable insight into the mechanisms by which SMLEs redistribute heat between open and ice-covered regions. However, the presentation could be improved by emphasizing the causal interpretation and quantitative comparisons more clearly. For instance, the roles of MHT and Qnet are described in detail, but it would be helpful to explicitly highlight how the presence of eddies amplifies meridional heat transport compared to the 2D simulation, and how this relates to changes in ice melt

or ML warming. Additionally, the text could more clearly distinguish between contributions from shortwave, longwave, and sensible fluxes in both regions, linking them directly to the eddy-induced heat redistribution. This would strengthen the physical interpretation and make the connection between eddies and observed heat budget changes more immediate for the reader.

The Antarctic results (Section 3.3) are presented clearly, with useful comparisons to Arctic simulations, but the section could benefit from emphasizing the physical interpretation of the differences. For example, the text could more explicitly link the faster ML deepening and weaker stratification in Antarctic winter to the smaller eddy impact on sea ice formation, and clarify why the summer eddy impacts are relatively insensitive to initial stratification. Including brief quantitative comparisons or ratios directly in the text (e.g., differences in MLD deepening or lateral density gradients) would help readers quickly grasp the relative magnitudes. Finally, a short discussion of the potential effects of neglected wind forcing on Antarctic results would strengthen the assessment of model limitations.

**Technical Corrections**

In Section 2, "Materials and Methods," it is recommended to split the content into two subsections for clarity: 2.1 "Model setup" and 2.2 "Residual-mean framework." This would improve the organization and make it easier for readers to follow the methods.

---

## Referee Comment (RC2)

**Referee comment on "Seasonal impact of submesoscale eddies on the ocean heat budget near the sea ice edge"**

Greig & Ferreira is a well-written and topical study that will be a useful contribution to the community. They show that submesoscale mixed layer eddies play a key role in the heat and sea ice budgets of the marginal ice zone, and investigate the sensitivity to background stratification and seasonal forcing. I think there are many interesting results, however I find the overall framing of the Arctic/Antarctic conditions to be a bit strange. The only difference between the "Arctic" and "Antarctic" simulations is the initial stratification. So why not just frame the study as testing the sensitivity to background stratification?

There are so many differences between the Arctic and Antarctic marginal ice zones in the real world: the forcing, large-scale circulation, winds, ice type/thickness, etc. Of course, these things are not represented in your idealized setup, which is fine. But I just think it's a bit misleading to present these simulations as representing an Arctic/Antarctic contrast. If this is your intent, then why choose these specific locations from EN4 to initialize the model (there can be very significant regional differences in the mean density structure within the Arctic and Antarctic, respectively)? My point is just that I think the results would be more accurately framed as testing the sensitivity to background stratification. This is a very valid and useful thing to do, and it's great to mention how this links broadly to the Arctic versus Antarctic, but I think this could be done with a bit more nuance (and I think these results are actually useful outside the context of comparing the Arctic to the Antarctic).

Also, to clarify, you are using the standard MITgcm sea ice package without any representation of sea ice floes? I feel this needs to be addressed directly. Sea ice floes in the marginal ice zone have similar length scales to submesoscale ocean flows, so most idealized modeling studies that deal with submesoscale sea ice-ocean interactions use either a simplistic representation of floes or a discrete element sea ice model. And typically, the relevant dynamics that are identified relate explicitly to floe-flow or floe-floe interactions. This needs to be discussed as it is important to understanding how your results relate to previous work. All this said, I want to emphasize that I think these are interesting simulations and it's a very nice paper dealing with an important subject! More detailed comments are provided below:

**Detailed comments:**

**Line 22-26:** Perhaps also worth mentioning that ocean heat has been invoked as a leading driver of the recent, rapid Antarctic sea ice loss (e.g. Purich & Doddridge, 2023).

Purich, A., E.W. Doddridge (2023). Record low Antarctic sea ice coverage indicates a new sea ice state. *Communications Earth & Environment*, **4**, 314.

**Line 46:** You are citing the pre-print rather than the actual published paper for Giddy et al. (the year should be 2021, not 2020).

Giddy, I., et al. (2021). Stirring of sea-ice meltwater enhances submesoscale fronts in the Southern Ocean. *Journal of Geophysical Research: Oceans*, **126**, e2020JC016814.

**Line 51 (and throughout):** Need to be clearer about whether you're referring to lateral or vertical heat transport by SMLEs.

Lines 70-71: Yes, I agree that more studies on submesoscale sea ice-ocean interactions have focused on mechanical effects. But Horvat et al. (2016) highlighted thermodynamic melt of sea ice floes by submesoscale eddies, and Gupta & Thompson (2022) considered both thermodynamic and mechanical interactions. These papers are cited elsewhere in the manuscript, but should at least be acknowledged. Another highly relevant paper to this work, which is not cited, is Brenner et al. (2023).

Brenner, S., et al. (2023). Scale-dependent air-sea exchange in the polar oceans: Floe-floe coupling in the generation of ice-ocean boundary layer turbulence. *Geophysical Research Letters*, **50**, e2023GL105703.

**Introduction:** a general comment on the introduction is that you do not do much to distinguish between the Arctic and Antarctic. Yet, a major focus of the simulations is comparing Arctic and Antarctic conditions. As I mentioned at the start, I'm a bit hesitant about this framing. But if you are going to proceed with this, then you should at least outline some of the differences between the poles in the introduction (and this could still be discussed briefly even if you do alter the framing around background stratification).

**Lines 87-101:** Since the model setup is essentially the same as Horvat et al. (2016) and much of this paragraph is just paraphrased from that paper, I think it is fine to condense this a bit and cite their paper for the details (or move to an appendix).

**Lines 107-110:** See my initial comments about the framing of these simulations as Arctic versus Antarctic.

**Line 113:** Does ERA5 have MLD as an output?

**Line 119:** How are the surface fluxes computed in the sea ice-covered part of the domain?

**Section 2.1:** First off, this could be Section 2.2 and you could have the earlier part of the Methods section be "Section 2.1 Model Set-up." But more importantly, this section needs to be introduced better. You need to start by stating what you are doing and why. i.e. you are using the isopycnal streamfunction and EOS to isolate the eddy-induced circulation, etc. Otherwise the section begins very abruptly and it's not even clear how this relates to the aims of the study until the end of subsection. Also, this technique is pretty widely used to diagnose eddy transports so you can probably condense some of the detail and just cite past work (e.g. Abernathey et al. 2011) or move to an appendix.

**Lines 186-187:** This is interesting, and what does this mean for the heat transport? If the front is salinity-dominated then the sign of the buoyancy flux is not necessarily the same as the sign of the heat flux.

**Equation 5 & Lines 219-221:** As I asked before, how are the surface fluxes computed in the ice-covered part of the domain? This is important to interpreting the heat budget.

**Lines 242-243:** The main narrative thread has gotten a bit lost by this point. So just to clarify, the 3-D simulation has a lower sea ice volume due to the meridional heat transport by SMLEs (rather than the vertical heat transport)? I think it's important to clarify lateral versus vertical fluxes.

**Lines 267-268:** I think this is important, lateral temperature gradients are weak because the temperature is near the freezing point, so you can have eddy buoyancy fluxes that are not accompanied by significant heat fluxes.

Line 331: "temporal" changes sounds more natural than "time" changes to me.

Lines 351-353: This gets at my initial comment about framing this as Arctic versus Antarctic. If the initial stratification is similar in summer at both poles, then these simulations will show similar results. But in the real world, there are many things that could contribute to differences in SMLE activity between the poles (forcing, winds, ice characteristics, topographic constraints, etc). In particular, the winds are very different. I know these things aren't represented in your simulations, but it seems like you're stating that the effect of SMLEs is the same at both poles in summer, and I think this is misleading. You address this a bit in lines 359-361, which is great. But I think you could expand upon this. Winds are known to be much stronger in the Antarctic, so you could state this directly, or even speculate on how this might manifest in the SMLE story.

**Lines 379-381:** This is interesting. If we assume that the APE reservoir scales with the MLD and lateral buoyancy gradient, then we might expect the eddy-induced overturning to be greater in the "Antarctic" i.e. deeper MLD case. So why is streamfunction weaker in the Antarctic setup? Is the lateral buoyancy gradient weaker?

Lines 418-450: I think it's really nice to list the key outcomes, but these are a bit dense. I think you should include all of this information in the Conclusions section, but you could consider putting some of this in the text (rather than bullet points) and really consolidating your key bullet points to 3-4 streamlined takeaways. I also feel you need to more directly highlight the lateral versus vertical transports when discussing the eddy heat fluxes. Is it correct to say that the eddy heat transport is primarily lateral in summer and vertical in winter? Or is the meridional heat transport just weaker in winter compared to summer (but still larger than the vertical fluxes)?

**Lines 456-467:** These seems a slightly odd note to end on. I agree that we need SMLE parametrizations that are designed for (or optimized for) the polar regions. But this work has

not dealt directly with parameterizations. I think it's useful to speculate about the implications of the results for parameterization development, but this would make more sense earlier on in this section rather than as the final sentences. In my opinion, it's more impactful to end with a strong statement about the significance of this work, rather than invoking some other unpublished work that the reader does not have access to.

**Lines 470-474:** It's a bit random to me to have such a small appendix given how much detail you gave about the model setup in the main text. If you're going to keep all of that detail in the Methods section, then you may as well just include these two sentences there. Or alternatively, you could move a more significant amount of those details here and streamline the Methods section in the main text.